# Sleep Apnea and Cardiovascular Risk in Patients with Prediabetes and Type 2 Diabetes

**DOI:** 10.3390/nu14234989

**Published:** 2022-11-24

**Authors:** Stavroula A. Paschou, Evanthia Bletsa, Katerina Saltiki, Paraskevi Kazakou, Kanella Kantreva, Paraskevi Katsaounou, Nikoletta Rovina, Georgia Trakada, Petros Bakakos, Charalambos V. Vlachopoulos, Theodora Psaltopoulou

**Affiliations:** 1Endocrine Unit and Diabetes Center, Department of Clinical Therapeutics, Alexandra Hospital, School of Medicine, National and Kapodistrian University of Athens, 11528 Athens, Greece; 23rd Department of Cardiology, Sotiria Hospital, School of Medicine, National and Kapodistrian University of Athens, 11527 Athens, Greece; 31st Department of Critical Care Medicine, Evangelismos Hospital, School of Medicine, National and Kapodistrian University of Athens, 10676 Athens, Greece; 41st Department of Respiratory Medicine, Sotiria Hospital, School of Medicine, National and Kapodistrian University of Athens, 11527 Athens, Greece; 5Respiratory Medicine Unit, Department of Clinical Therapeutics, Alexandra Hospital, School of Medicine, National and Kapodistrian University of Athens, 11528 Athens, Greece; 61st Department of Cardiology, Hippokration Hospital, School of Medicine, National and Kapodistrian University of Athens, 11527 Athens, Greece

**Keywords:** obstructive sleep apnea, prediabetes, diabetes, cardiovascular disease, vascular dysfunction, inflammation, oxidative stress

## Abstract

Obstructive sleep apnea (OSA) is a common but largely undiagnosed clinical condition, which is turning into a serious public health issue. Of note is that its prevalence is gradually increasing in parallel with the obesity and type 2 diabetes mellitus (T2DM) epidemics. The aim of this article is to comprehensively review the literature in order to evaluate the cardiovascular (CV) risk among patients with OSA and prediabetes or T2DM. OSA seems to be an independent risk factor for the development as well as the progression of T2DM, whereas it is associated with T2DM-related macrovascular and microvascular complications. OSA may also act as a potential risk factor for the presentation and development of CV disease, such as hypertension, coronary artery disease, heart failure, pulmonary hypertension, atrial fibrillation and other cardiac arrythmias, as well as stroke. OSA and T2DM also share common pathophysiological mechanisms leading to atherosclerosis. Considering that the coexistence of OSA and T2DM is an independent and cumulative risk factor for CV mortality, more so than the two diseases separately, clinicians and healthcare professionals should be aware of and screen for OSA in patients with T2DM. Notably, targeted therapy for both conditions seems to substantially improve CV prognosis.

## 1. OSA: Prevalence, Diagnosis, Pathophysiology, and Treatment

Obstructive sleep apnea (OSA) is a common but largely undiagnosed clinical condition, which is turning into a serious public health issue. It affects almost 1 billion people worldwide, and more than 30 million people are underdiagnosed in Europe [1,2]. Of note is that its prevalence is gradually increasing in parallel with the obesity and diabetes epidemics [3]. OSA is characterized by autonomous nervous system disturbances and the recurrent complete or partial collapse of upper airways (apneas or hyponeas) during sleep, leading to intrathoracic pressure changes, periodic reduction, or cessations in airflow; as a result, hypoxia, hypercapnia and frequent arousals from sleep occur [3,4].

Mechanisms involved in pharyngeal collapses are quite complicated and multifactorial, and are highly associated with instability of the upper airway. Typical symptoms include gasping, witnessed apneas, nocturia, headache, intense daytime sleepiness, depression, anxiety or irritation, and deficits in memory and attention, with the subsequent risk of road accident by day [5,6,7,8,9]. This clinical condition of altered sleep architecture leads to impaired performance and seriously affected quality of life, thus negatively impacting both physical and mental health [10]. Obesity seems to be the most prevalent risk factor for OSA, with more than 40% of patients with body mass index (BMI) ≥ 30 kg/m^2^ suffering from OSA [4]. Notably, OSA is much more widespread among patients with cardiovascular risk factors such as type 2 diabetes mellitus (T2DM), hypertension, or established cardiovascular disease (CVD) [11], with its prevalence ranging from 30% to 60% among these patients [12].

Nocturnal polysomnography is the gold standard for diagnosing OSA [13]. Apnea hypopnea index (AHI) seems to be a strong indicator of OSA severity, as well as a valuable clinical marker of whether or not to treat OSA. According to the American Academy of Sleep Medicine (AASM), OSA can be classified according to the AHI, which defines mild OSA as 10–15 events/h, moderate OSA as 15–30 events/h, and severe OSA as ≥30 events/h [14]. However, this diagnostic procedure cannot be applied in all patients at risk for sleep apnea, considering that it is quite expensive, time-demanding and inconvenient for some patients. Thus, AASM recommendations for the diagnosis of OSA also include questionnaires and other prediction algorithms, along with home sleep apnea testing by respiratory polygraphy [14]. So far, there are no pharmacological therapies approved for the management of OSA [4]. Nevertheless, AD109, a combination of the selective norepinephrine reuptake inhibitor, atomoxetine, with the selective antimuscarinic, aroxybutynin, which has completed the Phase II clinical trials, activates the upper airway dilator muscles, maintains an open airway during sleep, and thus seems to be a promising and effective novel oral medication for OSA [15]. Although continuous positive airway pressure (CPAP) seems to be quite a beneficial therapy for the management of OSA, a large meta-analysis indicated unexpectedly that CPAP does not improve cardiovascular outcomes among patients with OSA [16], possibly due to lack of adherence among these populations [17]. Nevertheless, other available conservative strategies include lifestyle intervention programs, such as weight loss, treating the underlying metabolic disorder, and CVD seem to be quite effective, offering successful management of OSA in more than 50% of the patients [18,19].

The aim of this article is to evaluate the cardiovascular risk (CV) in patients with OSA and prediabetes or T2DM. More specifically, we comprehensively reviewed the literature and present: (1) the relationship between OSA and metabolic disorders, (2) the relationship between OSA and CVD, (3) the CV risk among patients with both OSA and prediabetes or T2DM.

## 2. Methods

To identify eligible studies, a literature search was conducted in electronic databases, mainly PubMed, from inception until October 2022, using combinations of the key terms “obstructive sleep apnea”, “prediabetes”, “type 2 diabetes” and “cardiovascular disease.”

The inclusion criteria, established prior to the literature search, were English language papers of human studies investigating the association of OSA with metabolic disorders, the association of OSA with CVD, or the CV risk among patients with both OSA and prediabetes or T2DM. The scientific data were collected, analyzed, and qualitatively re-synthesized.

## 3. The Relationship between Sleep Apnea and Metabolic Disorders

T2DΜ is a systemic disease with detrimental macro and microvascular complications. The incidence of T2DΜ is rising alarmingly, with the global statistics to demonstrate that patients suffering from the disease will raise from 451 million to 693 up to 2045 [20]. OSA appears to be associated with an increased risk for the development of obesity, lipid profile derangements, non-alcoholic fatty liver disease (NAFLD), gout, as well as prediabetes and T2DM [21]. Obesity seems to be a major risk factor for the development of OSA, considering that up to 60% of obese patients suffer from OSA [22]. Similarly, the prevalence of OSA among patients with T2DM ranges from 55% to 86% [23], while patients with T2DM are at higher risk of developing OSA when compared to patients without diabetes [24,25].

According to the current literature, OSA seems to be an independent risk factor for the development of T2DM [26,27,28]. Interestingly, the results of three large-scale population-based cohort studies including more than 145,000 participants demonstrated that the presence of OSA increases the risk of developing T2DM by 37%, independently of demographic, lifestyle and anthropometric factors, while insulin-treated T2DM is strongly correlated with a 43% greater risk of OSA, particularly in women, possibly due to long duration and insufficient glycemic control [29]. According to a systematic review and meta-analysis, OSA is not only a risk factor for the new occurrence of T2DM, but there is also a positive relationship between the severity of OSA and the risk of T2DM. More specifically, mild dose AHI increases the risk of T2DM by 23%, while severe-dose AHI raises this risk with an even higher odds ratio [30].

Notably, there is evidence that the development and progression of OSA are strongly associated with the deterioration of glycemic status [31,32,33]. Moreover, among patients with T2DM, OSA is strongly associated with microvascular complications, such as peripheral neuropathy [34], peripheral arterial disease [35], diabetic nephropathy [36] and pre-proliferative/proliferative diabetic retinopathy [37] in longitudinal studies. Of note is that patients with T2DM and moderate-to-severe OSA are 3.05 times more likely to have any T2DM-related complications when compared to those with mild or no OSA, and this relationship is mainly mediated by the presence of hypertension [38].

OSA and T2DM seem to have a bidirectional relationship. Intermittent hypoxemia, autonomous nervous system hyperactivity, and sleep fragmentation resulting from OSA are the main contributors to the development of metabolic dysregulation, characterized by insulin resistance, glucose intolerance, b-cell dysfunction and T2DM [23,39,40]. To explain this in more detail, intermittent hypoxia and repeated arousals in OSA negatively affect autonomous nervous system function, leading to the secretion of catecholamines. The increase in epinephrine, norepinephrine, and cortisol secretion contributes further to impaired glucose metabolism and insulin sensitivity, as well as b-cell dysfunction due to increased gluconeogenesis and decreased glucose uptake [40,41]. Conversely, pre-existing T2DM provokes abnormalities of ventilatory and upper airway neural control and leads to peripheral neuropathy, thus accelerating the progression of OSA [39,42]. Furthermore, abnormalities in autonomous nervous system activity, activation of oxidative stress, and inflammatory pathways observed in T2DM may provoke further sleep-disordered breathing [23,39].

Furthermore, intermittent hypoxia causes hyperactivation of the hypothalamic pituitary-adrenal axis activation and adipokine dysregulation, leading to fat accumulation and obesity [43,44,45]. Leptin is produced by adipose tissue as an appetite suppressant and respiratory stimulant [46]. Increased leptin levels along with leptin receptor resistance, a common metabolic disorder in T2DM, provoke further metabolic and respiratory disturbances [47]. Moreover, it seems to cause neuromechanical dysfunction of the upper airway muscles, increasing pharyngeal collapsibility during sleep [48]. Additionally, lipid metabolism disorders, including hypercholesterolemia, triglyceridemia, and lipide peroxidation have been reported among patients with OSA [49,50]. Moreover, low levels of high-density lipoprotein (HDL) cholesterol and high levels of low-density lipoprotein (LDL) cholesterol have been also reported among patients with moderate/severe OSA [51].

## 4. The Relationship between Sleep Apnea and CVD

OSA is a quite a prevalent comorbidity among patients with CVD, considering that 38 to 65% of patients with coronary artery disease (CAD) and 12 to 55% of patients with heart failure (HF) may have co-existing OSA across various ethnicities [52]. Moreover, high rates of sudden cardiac death during sleep have been reported among patients with OSA [53]. Several studies highlight that OSA may act as a protentional risk factor for the occurrence and development of CVD, such as hypertension [54,55], CAD [56], HF [57,58], pulmonary hypertension [53,59], atrial fibrillation (AF) and other cardiac arrythmias [60,61], as well as stroke [62,63].

OSA and hypertension frequently co-exist. Notably, the prevalence of OSA ranges from 30% to 50% among hypertensive patients [64]. Patients with OSA are at higher risk of developing hypertension within 4 years of OSA diagnosis, independently of known co-existing risk factors [65]. Notably, patients with hypertension and OSA display specific hypertension-associated clinical patterns, such as resistant hypertension, masked hypertension, nondripping nocturnal blood pressure [66]. Indeed, some patients may have increased blood pressure during sleep only, or during sleep and wakefulness, thus a 24-h monitoring of blood pressure is highly advised among these patients [67].

Notably, individuals with sleep-disordered breathing, especially severe OSA, have a two- to fourfold higher risk of complex arrhythmias than those without sleep-disordered breathing [68]. Moreover, higher rates of major adverse cardiac events (MACEs) have been reported among patients with moderate-severe OSA and AF [69]. Furthermore, patients with OSA and AF are more likely to fail antiarrhythmic therapy [70]. These data imply that there is a strong relationship between the presence of AF, the severity of OSA, and the risk of cardiovascular events.

Considering that OSA has been linked to AF, it seems to be an independent risk factor for ischemic stroke, as well as stroke reoccurrence [71]. Severity of OSA is also positively correlated with the incidence of stroke [72]. Notably, it has been reported that patients with severe OSA have a two-fold higher risk of stroke within 6 years of diagnosis when compared to those without [73]. Moreover, patients with stroke and OSA are more likely to experience adverse neurocognitive outcomes, such as delayed functional recovery and motor recovery, daytime somnolence, depression, and longer periods of hospitalization in the neurorehabilitation units when compared to those without OSA [74].

The risk of fatal and non-fatal CV events among patients with OSA is up to 3 times higher when compared to the controls [75]. According to a meta-analysis, OSA is a significant predictor of serious adverse outcomes among patients with CVD or cerebrovascular disease [76]. More specifically, OSA was significantly associated with increased risk of stroke (RR 1.94, 95%, 1.29–2.92), CVD (RR 1.83, 95% CI, 1.15–2.93), and all-cause mortality (RR 1.59, 95% CI, 1.33–1.89) after stroke or CVD. Moreover, Salari et al. reported also OSA as a significant CV risk factor; moreover, there is a significant relationship between the severity of OSA and the risk of CVD, heart attack and CV death, according to their recent meta-analysis [77]. According to current literature, OSA is an independent predictor of MACEs and cerebrovascular events, as well as CV mortality among patients undergoing post-percutaneous coronary intervention (PCI), with an adjusted hazard ratio of 1.57 (95% CI 1.10–2.24) [78]. Furthermore, it has been indicated that patients hospitalized with non-ST elevation acute coronary syndrome (ACS) with an history of OSA have a greater risk of composite of MACEs, such as death, non-fatal MI and refractory angina during hospitalization, which significantly aggravates their CV prognosis [79].

OSA seems to act as a risk factor for the development of both HF with reduced eject fraction (HFrEF) and HF with preserved eject fraction (HFpEF) [80]. The prevalence of OSA is around 10–35% among patients with HF [57]. Of note is that observational studies demonstrate that survival rates are generally lower among patients with HF and OSA compared to those without OSA [57].

According to a recent meta-analysis, OSA has been also associated with an overall significant increase in risk of aortic dissection by 60%, possibly due to the mechanisms of sympathetic vaso-activity and inflammation, induced by intermittent hypoxia [81]. Additionally, it was also reported that patients with moderate or severe OSA have a greater risk of aortic dissection by up to 443%, suggesting that the severity of OSA has a positive correlation with the risk of aortic dissection [81].

OSA triggers respiratory, nervous, metabolic, and immune system activation, thus raising the risk of CVD. OSA-induced intermittent hypoxia and sleep disruption promote autonomic nervous system dysfunction, increased nocturnal sympathetic activity and catecholamine secretion, activation of the renin-angiotensin-aldosterone system, oxidative stress, and low-grade chronic inflammation [82,83,84]. The hemodynamic consequences of OSA also include severe vascular dysfunction and remodeling, such as endothelial dysfunction and arterial stiffness, vascular inflammation, thrombosis and platelet reactivity, leading to accelerated atherosclerosis [85,86,87]. Notably, endothelial dysfunction and arterial stiffness, strong markers of atherosclerosis [88,89,90], are strongly associated with OSA, mediated by hypoxia, hypercapnia, and oxidative stress, thus increasing CV risk [91]. Impaired endothelium-dependent vasodilation along with suppressed production of endothelium-dependent vasodilator substances, such as nitric oxide (NO), and elevated levels of endothelin-1 and other growth factors, have been described in OSA [92,93]. Moreover, OSA has been associated with hypercoagulability and platelet reactivity, since elevated levels of fibrinogen and other prothrombotic factors have been reported among patients with OSA [94].

Notably, OSA-induced intermittent hypoxemia stimulates ventilation, thus leading to increased negative pressure in the chest, increased resistance, heart rate, and blood pressure. As a result, preload and afterload are significantly elevated, particularly among patients with established CVD, leading to left ventricular hypertrophy, and atrial stretch and enlargement, as well as myocardial remodeling and fibrosis [95], which act as a good substrate for the occurrence of AF [96]. Moreover, a greater left ventricular afterload may provoke myocardial ischemia, QTc prolongation, and ventricular arrhythmias [97]. Moreover, repetitive episodes of apnea during sleep activate the sympathoadrenal system, resulting in a sustained elevation of sympathetic activity and increased concentrations of catecholamines while awake, which are a good substrate for ventricular repolarization and the occurrence of arrhythmogenic events [98,99].

Based on the previous findings, patients with OSA are more likely to develop coronary atherosclerosis, myocardial injury, left ventricular remodeling, and vascular dysfunction. According to a large meta-analysis, patients with OSA have significantly higher levels of carotid-intima media thickness (CIMT), which is a strong marker of atherosclerotic process [100]. OSA has also been associated with coronary artery calcification, plaque instability and vulnerability [56,101].

Notably, common echocardiographic findings among patients with OSA are the following: elevated right-side pressures, pulmonary hypertension, tricuspid and mitral regurgitation, diastolic dysfunction, left atrial enlargements, and left ventricular hypertrophy [102]. According to a recent meta-analysis of imaging studies, including 3082 patients with OSA and 1774 matched controls, higher left atrial diameter, higher left atrium volume index, wider left ventricular end-systolic diameter, left ventricular end-diastolic diameter, and left ventricular mass, higher left ventricular mass index, interventricular septum diameter and posterior wall diameter, as well as higher left ventricular myocardial performance index were observed among patients with OSA [103]. In addition, left ventricular ejection fraction was significantly decreased in OSA patients, while increased right ventricular diameter and right ventricular myocardial performance index along with decreased tricuspid annular plane systolic excursion and right ventricular fractional area change were displayed among patients with OSA [103].

## 5. CVD Risk among Patients with Sleep Apnea and Prediabetes or T2DM

Evidence reveals that both T2DM and OSA are independent risk factors for the development and the progression of CVD. Indeed, patients with OSA and T2DM are more likely to develop CAD and HF when compared to those diagnosed only with T2DM [104]. Interestingly, risk of CVD and mortality rates are even more elevated if T2DM is diagnosed before OSA [105]. Therefore, patients with T2DM who develop OSA are a high-risk population for CVD, and clinicians should be aware of this bidirectional association and the CV complications, which are about to be further discussed. The main studies evaluating the CV risk among patients with OSA and prediabetes or T2DM are presented in Table 1.

Interestingly, OSA among patients with T2DM significantly increases the risk of developing CVD, as well as the respective rates of mortality. According to a population-based cohort study of 3667 patients and 10,450 matched controls, patients who developed OSA after their diagnosis of T2DM were at higher risk of developing ischemic CAD by 55%, HF by 67%, and stroke or transient ischemic attack (TIA) by 57%, when compared to patients with T2DM and no OSA [106]. As a result, all-cause mortality was detrimentally increased by 24%. In the secondary analysis exploring outcomes in participants with T2DM and prevalent OSA compared with T2DM only, patients with OSA continued to be at increased risk of composite CVD [106].

There is evidence that both the presence and severity of OSA seem to increase the risk of cardiac arrhythmias among patients with T2DM. Patients with T2DM who developed OSA are at higher risk of developing AF by 53% compared with patients without diagnosed OSA [106]. Additionally, a recent cross-sectional study revealed that the presence and severity of OSA are strongly correlated with a 2.3-fold increased risk of QTc prolongation among patients with T2DM [97]. Of note, among patients with T2DM and QTc ≥ 418 ms, older patients (OR: 1.042, 95% CI: 1.042–1.064, *p* < 0.001), females (OR: 2.36, 95% CI: 1.371–4.063, *p* < 0.01), and patients with higher BMI (OR: 1.113, 95% CI: 1.037–1.195, *p* < 0.01) were significantly more likely to have OSA. These data provide evidence that OSA-related QTc prolongation may be an additional risk factor for CVD in patients with T2DM, highlighting the importance of evaluating global CV screening for arrhythmia risk stratification [113].

According to the current literature, OSA seems to be an independent risk factor not only for T2DM, but also for T2DM-related macrovascular and microvascular complications. According to a population-based cohort study of 3667 patients and 10,450 matched controls, patients with T2DM who developed OSA are at higher risk of developing peripheral neuropathy by 32%, diabetic foot disease by 42%, chronic kidney disease by 18%, and albuminuria by 11% when compared to patients with T2DM and no OSA [106]. In a longitudinal, large-scale population-based cohort study in Finland, including 36,963 participants, it was indicated that patients with OSA face a 36% greater risk of CVD, after a follow-up of 25 years, than the general population [107]. Notably, this risk was even prominent among women, who usually remain undiagnosed and untreated. Furthermore, patients with OSA had a 1,75-fold increased risk of diabetic kidney disease [107]. In particular, patients with both T2DM and OSA had an even higher risk of CVD (by 36%) and diabetic kidney disease, and, therefore, a markedly increased all-cause mortality when compared to those with T2DM and no OSA [107].

Moreover, T2DM seems to increase the risk of MACEs by 64%, as well as hospitalization for unstable angina and all-end events among patients with OSA [108]. In this cohort study, patients with OSA and T2DM experienced more end-events after a median follow-up of 42 months when compared to patients without T2DM. Interestingly, overweight and obese females ≥ 70 years with T2DM and mild OSA present a substantially higher risk of MACEs. Furthermore, the SantOSA cohort, a prospective cohort study of 1447 patients, verifies that the coexistence of OSA and T2DM is a greater independent risk factor for CV mortality than either of the two diseases in isolation. Moreover, OSA in combination with T2DM at baseline was associated with a 3,4-fold higher prevalence of CVD, as well as a 2,3-fold increased CV mortality after a median follow-up of 5 years [109].

Similarly, OSA is associated with a 2.5-fold increased risk of 1-year MACEs among patients with T2DM following ACS, but not in patients without T2DM [110]. Combined OSA and longer hypoxia duration, defined as time with arterial oxygen saturation < 90% for more than 22 min, leads to a further risk in the MACEs by 31.0% in patients with T2DM. Moreover, there is evidence that the presence of OSA among patients with T2DM post-percutaneous coronary intervention (PCI) leads to adverse outcomes, worsening patients’ prognoses. Moreover, it has been reported that OSA is associated with a 2-fold risk in MACEs and cerebrovascular events in patients with T2DM after 3 years of successful PCI, but not in those with no T2DM [111]. Indeed, ACS leading to adverse CV outcomes is mainly provoked by new lesions of thin cap fibroatheromas in non-culprit vessels among patients with T2DM [114]. Similarly, severe and untreated OSA is strongly correlated with high rates of repeated revascularizations after PCI, mainly to non-target lesions [115].

The increased risk of MACEs and all-cause mortality among OSA patients with T2DM is also verified by Wang et al. According to a recent meta-analysis of 1168 patients with T2DM following PCI (of whom 614 had co-existing OSA), MACEs and all-cause mortality were significantly higher in the OSA subgroup [112]. Therefore, patients with T2DM and co-existing OSA face a greater risk for adverse CV outcomes after PCI when compared to those with T2DM and no OSA, thus requiring close monitoring and strict management strategies. Notably, the increased risk of CV events is predominately reported in patients with OSA and poor glycemic control at baseline [112]. Therefore, the co-existence of OSA and T2DM seems to have a synergistic effect, thus promoting the development and progression of atherosclerosis, as well as increasing the risk of CV events among these patients.

These results provide evidence that OSA and T2DM share common mechanisms of atherosclerosis, leading to serious CV events, as presented in Figure 1. OSA-induced repeated hypoxemia accompanied by reoxygenation has been associated with low-grade chronic systemic inflammation, oxidative stress, and endothelial dysfunction, thus triggering the development of T2DM and the related macrovascular and microvascular complications. Specifically, the inflammatory profile of OSA is characterized by the recruitment of macrophages and the presence of circulating inflammatory biomarkers, such as C-Reactive Protein (CRP), chemokines and cytokines such as Interleukin 6 (IL-6), tumor necrosis factor α (TNF-α), nuclear factor kappa B (NFK-B), as well as adhesions molecules such as selectins, intracellular adhesion molecule-1 (ICAM-1), and vascular intracellular adhesion molecule-1 (VCAM-1) [116,117]. Furthermore, OSA may disrupt the balance between pro-oxidant and antioxidant factors. High levels of superoxide anions in circulating leukocytes, production of ROS, increase in lipid peroxidation, and decrease in antioxidant defense have been reported among patients with OSA, possibly due to the repetition of hypoxia-reoxygenation cycles [118,119]. Hypoxia inducible factor-1 (HIF-1), a major transcription factor, is activated by hypoxia, and when chronic, it negatively modifies the transcription of plenty genes, thus provoking detrimental consequences for CV system [120].

According to the current literature, CPAP seems to be a quite beneficial therapy for the management of OSA. Nevertheless, among patients with established CVD and OSA, CPAP therapy does not improve glycemic control among patients with prediabetes or T2DM over standard care treatment. Moreover, CPAP therapy did not modify serum glucose levels, hemoglobin A1C (hbA1c) or antidiabetic medication among these patients after a median follow up of 4.3 years [121]. Similarly, empagliflozin, a sodium glucose -cotransporter inhibitor −2 (SGLT-2i), seems to be quite promising [122,123,124]. In the EMPA-REG OUTCOME trial, 6% of the participants had OSA at baseline, and these patients faced higher rates of CV and renal events, as well as higher comorbidity during the trial [125]. Notably, empagliflozin improved cardiometabolic risk factors, as well as MACEs, hospitalization for HF, mortality, and renal outcomes, regardless of pre-existing OSA. It may also decrease the incidence of new-onset OSA, possibly due to weight loss and the favorable effect of visceral adiposity [126]. Moreover, metformin, the cornerstone in the management of T2DM, may improve sleep quality, although it does not modify the prevalence of OSA, according to retrospective cohort studies [127,128]. Despite that, these preliminary data remain weak and should be further examined and elucidated.

## 6. Conclusions

Current evidence indicates that both T2DM and OSA are independent risk factors for the development and the progression of CVD. These chronic conditions seem to share detrimental pathophysiological pathways, increasing the risk of CV complications and markedly worsening patient prognosis. This effect is especially profound and synergistic when these two clinical conditions co-exist. Thus, patients with T2DM who develop OSA are at higher CV risk, and as a result clinicians and healthcare professionals should be aware of OSA co-existence among patients with T2DM. Of note, targeted therapy for OSA and T2DM seems to substantially improve their CV prognosis. Nevertheless, OSA still remains undiagnosed and untreated in everyday clinical practice. Thereby, personalized medicine approach, careful assessment of co-morbidities, as well as better CV risk stratification and CVD prevention among patients with T2DM and OSA are of major importance.

## Figures and Tables

**Figure 1 nutrients-14-04989-f001:**
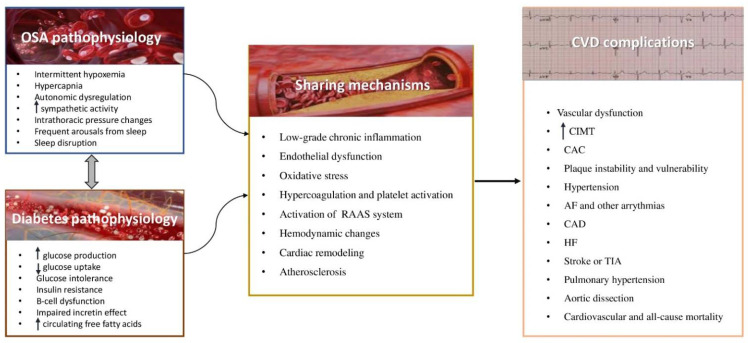
OSA and diabetes sharing pathophysiological mechanisms leading to CVD. OSA: obstructive sleep apnea. CVD: cardiovascular disease, RAAS: renin- angiotensin aldosterone system, CIMT: carotid intima media thickness, CAC: coronary artery calcification, AF: atrial fibrillation, CAD: coronary artery disease, HF: heart failure, TIA: transient ischemic attack.

**Table 1 nutrients-14-04989-t001:** Main studies evaluating the CV risk among patients with OSA and prediabetes or T2DM.

Author	Journal	Type of Study	Patients Characteristics	Main Findings
Adderley et al. [106]	Diabetes Care	Retrospective cohort study	▪ 3667 patients who developed OSA after their T2DM diagnosis and 10,450 matched control participants	▪ Patients with OSA and T2DM were at higher risk of CAD by 55%, AF by 53%, HF by 67%, as well as stroke or TIA by 57%, PN by 32%, DFD by 42%, CKD by 18% and albuminuria by 11%, and all-cause mortality by 24%.
Strausz et al. [107]	BMJ Open	Longitudinal study	▪ 36,963 participants (>500,000 person years) ▪ follow-up: 25 years	▪ Patients with OSA faced a 36% greater risk of CVD than the general population. ▪ OSA increased the risk of CVD by 36% and all-cause mortality by 35% among patients with T2DM.
Su et al. [108]	BMC Geriatr	Prospective cohort study	▪ 1113 elderly patients with OSA ▪ 23.9% of patients with co-existing T2DM ▪ follow-up: 42 months	▪ T2DM increased the risk of MACEs by 64%, as well as hospitalization for unstable angina and all-end events among patients with OSA. ▪ Overweight or obese females ≥70 years with T2DM and OSA presented a substantially higher risk of MACEs.
Labarca et al. [109]	Sleep Breath	Prospective cohort study	▪ 1447 patients, 151 with co-existing OSA and T2DM, 736 only with OSA, 141 only with T2DM, 441 controls ▪ follow-up: 5 years	▪ OSA in combination with T2DM at baseline was associated with a 3,4-fold higher prevalence of CVD as well as a 2,3-fold increase in cardiovascular mortality.
Wang et al. [110]	BMJ Open Diabetes Res Care	Prospective cohort study	▪ 804 patients following ACS ▪ 30.8% of patients with co-existing T2DM ▪ 50.1% of patients with co-existing OSA	▪ OSA was associated with 2.5-fold increased risk of 1 year MACEs among patients with T2DM following ACS. ▪ Combined OSA and longer hypoxia duration increased the risk of MACEs by 31.0% in these patients.
Koo et al. [111]	Diabetes Care	Observational cohort study	▪ 1311 patients undergoing PCI ▪ 20.7% of patients with co-existing OSA and T2DM, 24.6% only with OSA, 21.7% only with T2DM ▪ follow-up: 3 years	▪ OSA was associated with a 2-fold increased risk in MACEs and cerebrovascular events in patients with T2DM following successful PCI.
Wang et al. [112]	Diabetes Ther	Systematic review and meta-analysis	▪ 1168 patients with T2DM following PCI ▪ 52.5% of patients with co-existing OSA	▪ OSA was associated with a 2.2-fold increased risk of MACEs and a 1,9-fold increase in all-cause mortality among patients with T2DM.
Shi et al. [97]	Med Sci Monit	Cross-sectional study	▪ 358 patients with T2DM ▪ 59.2% of patients with co-existing OSA	▪ Presence and severity of OSA was strongly correlated with a 2,3-fold increased risk of QT_c_ prolongation among patients with T2DM. ▪ Among patients with T2DM and QTc ≥ 418 ms, older patients, females, and patients with higher BMI were more likely to have OSA.

OSA: obstructive sleep apnea, T2DM: type 2 diabetes mellitus, CAD: coronary artery disease, AF: atrial fibrillation, HF: heart failure, TIA: transient ischemic attack, PN: peripheral neuropathy, DFD: diabetic foot disease, CKD: chronic kidney disease, CVD: cardiovascular disease, MACEs: major adverse cardiac events, ACS: acute coronary syndrome, PCI: percutaneous coronary intervention, BMI: body mass index.

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
