# Peer review of "Sleep Apnea and Cardiovascular Risk in Patients with Prediabetes and Type 2 Diabetes"

_nutrients, 2022, doi:10.3390/nu14234989_

Round 1

Reviewer 1 Report

The current review discusses the literature on the association between obstructive sleep apnea (OSA) and the cardiovascular (CV) risk among patients with OSA and prediabetes or T2DM. In this context, authors provide a thorough literature review with interesting and important implications. The discussion is coherent, providing insight into the common pathophysiological mechanisms of the conditions  alongside clinically significant suggestions for the healthcare professionals. The figure also provides an insightful summary. Overall, it was a pleasure to review this highly interesting review.

Author Response

We thank the reviewer for the kind suggestions, which contributed to the improvement of the manuscript. All of them have been taken into consideration. 

Reviewer 2 Report

In this review article, Paschou et al. systematically summarized the literatures in studying correlations between the cardiovascular risk and the coexistence of obstructive sleep apnea and type 2 diabetes mellitus in patients. The authors carefully summarized statistical data from studies about the influence of OSA on T2DM and CVD respectively, and then the combination of OSA and T2DM on CV risk. They also discussed possible shared mechanism of OSA and T2DM in CVD complications and indicated the importance of this bidirectional association in future clinical research. The manuscript is in good quality and well structured, and sufficient discussions were provided to emphasize the combinatorial effect of OSA and T2DM. Thus, this manuscript is suitable for publication in Nutrients, and few minor issues could be further improved. 

In this manuscript, the authors compared the ratio increase when patients carry both OSA and T2DM. However, in a few cases it is not clearly described the increase is compared with health controls or patients only carry one type of risk factors. Also, it will be more informative to readers by summarizing both the percentage of increase and the baseline percentage in different control groups. A summarize of fold increase will be another way to show the data more comprehensively if the data are available from literatures.

Page 2 line 58, should specify which type of disease "this disorder" is referring to. 

Page 4 line 156 to 162, this paragraph is introducing OSA risks from the aspect of arrhythmias and AF, while the paragraphs above and below are focusing on hypertension. The coherence between this paragraph and its context is not optimal. This paragraph should be moved to later part of this section. 

Author Response

(The authors gave the same response as above.)

Reviewer 3 Report

The authors present important review findings regarding the impact of OSA on CVD risk in persons with DM and pre-DM.

The manuscript is organized and concise.

Suggestions:

Include a description of your online research search - databases, terms, inclusion/exclusion criteria and a PRISMA flow diagram.

https://guides.lib.unc.edu/systematic-reviews/write

Correct your use of semicolons - colons are the appropriate punctuation preceding lists.

Figure 1 needs a footnote with abbreviation definitions.

Correct the sentence on line 345.

Author Response

We thank the reviewers for the kind suggestions, which contributed to the improvement of the manuscript. All of them have been taken into consideration. 
